# Extrusion-Based 3D Printing of Pharmaceuticals—Evaluating Polymer (Sodium Alginate, HPC, HPMC)-Based Ink’s Suitability by Investigating Rheology

**DOI:** 10.3390/mi16020163

**Published:** 2025-01-30

**Authors:** Farzana Khan Rony, Georgia Kimbell, Toby R. Serrano, Destinee Clay, Shamsuddin Ilias, Mohammad A. Azad

**Affiliations:** 1Department of Applied Science and Technology, North Carolina A&T State University, Greensboro, NC 27411, USA; fkrony@aggies.ncat.edu; 2Materials Science and Process Engineering (MSPE) Lab, Department of Chemical, Biological, and Bioengineering, North Carolina A&T State University, Greensboro, NC 27411, USA; glkimbell@aggies.ncat.edu (G.K.); tserrano@aggies.ncat.edu (T.R.S.); djclay@aggies.ncat.edu (D.C.); 3Department of Chemical, Biological, and Bioengineering, North Carolina A&T State University, Greensboro, NC 27411, USA; ilias@ncat.edu

**Keywords:** extrusion-based 3D printing, pressure-assisted microsyringe (PAM) printing, rheology, polymer (sodium alginate, HPC, HPMC), pharmaceutical dosage (pill and film)

## Abstract

Three-dimensional printing is promising in the pharmaceutical industry for personalized medicine, on-demand production, tailored drug loading, etc. Pressure-assisted microsyringe (PAM) printing is popular due to its low cost, simple operation, and compatibility with heat-sensitive drugs but is limited by ink formulations lacking the essential characteristics, impacting their performance. This study evaluates inks based on sodium alginate (SA), hydroxypropyl cellulose (HPC H), and hydroxypropyl methylcellulose (HPMC K100 and K4) for PAM 3D printing by analyzing their rheology. The formulations included the model drug Fenofibrate, functional excipients (e.g., mannitol, polyethylene glycol, etc.), and water or water–ethanol mixtures. Pills and thin films as an oral dosage were printed using a 410 μm nozzle, a 10 mm/s speed, a 50% infill density, and a 60 kPa pressure. Among the various formulated inks, only the ink containing 0.8% SA achieved successful prints with the desired shape fidelity, linked to its rheological properties, which were assessed using flow, amplitude sweep, and thixotropy tests. This study concludes that (i) an ink’s rheological properties—viscosity, shear thinning, viscoelasticity, modulus, flow point, recovery, etc.—have to be considered to determine whether it will print well; (ii) printability is independent of the dosage form; and (iii) the optimal inks are viscoelastic solids with specific rheological traits. This research provides insights for developing polymer-based inks for effective PAM 3D printing in pharmaceuticals.

## 1. Introduction

Three-dimensional (3D) printing, also known as additive manufacturing, is the process of building a three-dimensional object by placing multiple successive layers of material [1,2,3]. Three-dimensional printing can be used to make custom-designed objects quickly, easily, and affordably [4]. To create an object using a 3D printer, a digital three-dimensional design of the object is created in a computer program such as SolidWorks, and this file is later used to instruct the printing device on how to construct the object [1,5]. In recent years, 3D printing has secured a foothold in the pharmaceutical industry due to its ability to customize drug products by adjusting how much drug is loaded, allowing for complex dosage forms, producing medicine on demand, and delivering pills containing multiple drugs or active pharmaceutical ingredients (APIs), etc., while the current manufacturing methods in the pharmaceutical industry, which employ a one-size-fits-all approach, are limited by scientific and technological constraints in achieving such capabilities [6,7,8,9]. Although there are many types of 3D printing, an extrusion-based 3D printing type known as pressure-assisted microsyringe (PAM) printing is gaining popularity in the pharmaceutical industry because it is simple, low-cost, and scalable and does not require heat (allowing for the use of heat-sensitive drugs) [1,4,6,10]. PAM printing involves the use of pressurized syringes to extrude polymer-based ink [11]. The PAM-type extrusion-based 3D printing process can be divided into three primary steps (shown in Figure 1): 1. ink preparation; 2. ink extrusion; and 3. ink deposition and 3D structure formation. Each step is essential to ensure acceptable printability with good reproducibility. Generally, the polymer and functional excipients are chosen to make sure they will provide the ink with suitable rheological properties, thus making the ink extrudable and giving it the ability to maintain the desired shape of the 3D-printed object [12]. The ink’s ability to flow through the print head nozzle and maintain its structure after 3D printing relies on the ink’s rheological properties, which are determined by parameters such as viscosity, viscoelasticity, recovery, etc. [4].

In general, 3D printing is performed based on trial and error, which is time-consuming and resource-intensive, especially for high-value products such as pharmaceuticals, regenerative medicine, etc. [13,14]. The difficulty with PAM 3D printing comes from the ability to develop polymer-based inks with a suitable rheology that can be extruded smoothly through the print head and hold their shape after being printed [15]. If the polymer-based ink does not have the proper rheological characteristics, it may create issues during the printing process, such as clogging the print head nozzle or defective printing [16,17]. It is also noted that suitable printing process parameters are essential to obtain printability and good reproducibility, and these parameters rely on the ink’s rheological properties [12,18]. Hence, developing a polymer-based ink suitable for PAM-type 3D printing is extremely challenging [17].

The polymers sodium alginate (SA), hydroxypropyl cellulose (HPC), and hydroxypropyl methylcellulose (HPMC) are commonly used in the pharmaceutical industry [19,20,21,22]. These polymers are classified as generally regarded as safe (GRAS) [23]. Recently, ink preparation using these polymers, sodium alginate [24,25], HPC [26,27], and HPMC [28,29], which are suitable for extrusion-based 3D printing, has been discussed. However, no study has evaluated all three of these polymers, compared the different rheological properties of these polymeric inks, and eventually connected them with the 3D printing outcomes for various oral dosages. This research study aims (i) to assess inks prepared using the polymers SA, HPC H, or HPMC K100 or K4 by investigating and comparing their rheology and connecting the rheological data with the ink’s applicability to PAM-type 3D printing and (ii) to observe whether there are any variations in the printing outcomes due to dosage variations such as pills and films, which are the most commonly used in an oral dosage form. The novelty of this research work relies on the exploration of these aims. To prepare the ink, Fenofibrate (FNB), used to treat high cholesterol levels, was used as the model drug, along with several functional excipients, including mannitol, polyethylene glycol (PEG), polyvinylpyrrolidone (PVP), and sodium dodecyl sulfate (SDS). Each of these excipients provides functionality and contributes to the ink’s overall rheological properties, i.e., its viscosity. It is noted that even though the functional excipients contribute to the rheology, this study focuses on the rheological variations caused by the polymers. Hence, the concentrations of the functional excipients were kept the same. Only the polymer’s concentration or type was varied. The ink’s rheological properties (i.e., viscosity, storage and loss modulus, viscoelasticity, flow behavior, recovery after shear, etc.) were evaluated using an Anton Paar MCR 302 rheometer (Graz, Austria). The ink was then used to 3D-print various dosage forms (i.e., pills, films) using a Cellink BioX PAM-type 3D printer.

## 2. Materials and Methods

### 2.1. Materials

The polymers sodium alginate (alginic acid sodium salt), HPC H, and HPMC (K100 (Methocel™ K100M Premium CR) or K4) were purchased from Acros Organics (Fair Lawn, NJ, USA), donated by Nisso America Inc. (New York, NY, USA), and donated by Colorcon^®^ (Harleysville, PA, USA), respectively. The molecular structures of these polymers are depicted in Figure 2. Here, sodium alginate and HPMC are biopolymers, while HPC is a synthetic derivative of the natural polymer cellulose. The model drug Fenofibrate was purchased from Tokyo Chemical Industry Co, Ltd., (Tokyo, Japan). The functional excipients D-mannitol 97%+, polyethylene glycol 6000 (PEG 6000), polyvinylpyrrolidone K 30 (PVP K30), and sodium dodecyl sulfate (SDS) were purchased from Alfa Aesar (Haverhill, MA, USA), Tokyo Chemical Industry Co, Ltd., (Tokyo, Japan), Alfa Aesar (Haverhill, MA, USA), and TCI America (Portland, OR, USA), respectively. It is noted that (i) mannitol was used as filler to increase the ink’s solid content and subsequently its viscosity [30], (ii) PEG was used as a plasticizer [31], (iii) PVP was used to improve the dispersion of the hydrophobic model drug FNB in the ink [32], and (iv) SDS is a surfactant and was used as a wetting agent [33]. Ethanol 89.5–91.5% (*v*/*v*), ACS-reagent-grade, was purchased from Ricca Chemical Company (Arlington, TX, USA).

### 2.2. Preparation of the Ink for 3D Printing

A 30 g batch size of printing ink (as a paste) was prepared using each of the polymers (SA, HPC H, and HPMC (K100 or K4)) investigated. A summary of the printing ink’s composition is shown in Table 1. The details of the ink preparation using each of these polymers are discussed below. It is noted that a brief preliminary study was carried out using different percentages of polymers (e.g., sodium alginate) to prepare the ink and its printability evaluated [36]. Subsequently, the polymer compositions were chosen for this study.

#### 2.2.1. Preparation of the SA Ink

Two ink formulations containing 0.8% and 1.6% (*w*/*w*) SA were developed. For the preparation of both SA inks, the specified weight (Table 1) of SA, FNB, PEG, mannitol, and PVP powders were ground using a mortar and pestle for 5 min and then added to deionized (DI) water. This mixture was put into a THINKY mixer (ARE-310, Tokyo, Japan) for 10 min at 2000 RPM to obtain a uniform paste.

#### 2.2.2. Preparation of the HPC H Ink

The HPC H ink was prepared in two steps. In the first step, an HPC gel was prepared following the procedure discussed in [34,37]. Initially, 8% and 12% (*w*/*w*) HPC H gels were prepared by gradually adding a small amount of a total of 4 g and 6g of HPC dry powder into 24 g and 36 g of DI water, respectively, which was heated to 50–60 °C and stirred using a magnetic stirrer bar at 400 rpm in a glass beaker. After adding all of the HPC, the mixture was stirred for another 5 min at 450 rpm, keeping the temperature the same. Then, the heating was stopped, and the mixture was cooled to 40 °C while stirring was continued. Once the mixture reached 40 °C, the remaining 22 g of DI (for 8%) or 8 g (for 12%) water was added. The mixture was further stirred for 10 min until the temperature reached 25 °C. This process formed a uniform gel with no visible particulates. In the second step, drug-loaded printing ink was prepared. To prepare the ink, the specified weight (in Table 1) of FNB, mannitol, PEG, and PVP powders were ground using a mortar and pestle for 5 min and then added to 12 g of the 8% and 12% HPC gel for the preparation of 3.2% HPC ink and 4.8% HPC ink. Then, the THINKY mixer (ARE-310) was used for 10 min at 2000 rpm, and a uniform paste was obtained.

#### 2.2.3. Preparation of the HPMC (K100 or K4) Ink

The HPMC-based ink was also prepared in two steps—gel and then ink. The 2% (*w*/*w*) HPMC K100 gel was prepared by mixing 1 g of HPMC K100 powder into 30 g of DI water at a temperature of 90 °C in a beaker using an overhead stirrer at 450 rpm until the mixture was uniform. The temperature to heat the water was chosen based on Khaled et al. [38]. Then, 19 g of DI water at a temperature of 40 °C was added to the mixture and stirred for 30 min until it had mixed uniformly. A similar procedure was used to prepare the 2% HPMC K4 gel. The gel was kept in a refrigerator at 4 °C for 24 h to remove any entrapped air bubbles. To prepare the printing ink, FNB and the other functional excipients at the specified weights mentioned in Table 1 were ground using a mortar and pestle for 5 min and then added to 15.75 g of the 2% HPMC K100 or K4 gel along with 5.25 g of ethanol. The THINKY mixer (ARE-310) was used for uniform mixing at 2000 rpm for 10 min. Ethanol was added to facilitate the dissolution of HPMC into the water.

### 2.3. Design of the Pills and Films

The printing process starts with the design of a 3D object in a CAD (Computer-Aided Design) program. The 3D CAD design file is then translated into a g.code (geometric code) using a 3D slicing software (i.e., Slic3r). G-code is a language that directs the 3D printer head on where to move, when to move, how to move, and how much material to extrude. In the end, the printer prints the structure based on the instructions from the g.code. In this study, SolidWorks (Dassault Systèms, 2018), a CAD program, was used to generate the 3D models of the pill and film (Figure 3). The pill was designed with a height of 5 mm and a diameter of 12 mm, and the film was designed with the following dimensions: 15 mm (length) × 10 mm (height) × 0.35 mm (thickness).

### 2.4. Rheology

Rheology is the science of how a material flows and deforms when it is acted upon by a force [39]. The rheology of each polymer-based ink was analyzed using an Anton Paar MCR 302 rheometer with a PP 25 measuring plate, a Peltier hood to control the temperature of the ink, and a 1 mm sample gap. Flow, amplitude sweep, and thixotropy tests were conducted using the rheometer. All of the tests were performed at 25 °C and a 1 Hz frequency. Flow tests were performed for inks containing SA, HPC, HPMC K100, or HPMC K4 using a shear rate of 0.1 to 1000 1/s. The viscosity curves were plotted using viscosity (mPa.s) vs. shear rate (1/s) data. From the viscosity curve, the shear thinning behavior of the ink was evaluated. An amplitude sweep oscillatory test was performed on each ink with a shear strain ranging between 0.001 and 1000%. The storage and loss modulus (Pa) were plotted against the shear strain (%) [40]. The amplitude sweep can report the yield point (stress), flow point (stress), flow transition index, and phase shift angle of the material [41]. The thixotropy test, a rotational test, consists of three intervals. The test measures the viscosity of the ink at all three intervals [42]. The 1st (known as rest) and 3rd (known as recovery) intervals consist of oscillation at a shear strain percentage within the linear viscoelastic (LVE) region, determined from the amplitude sweep test. The 1st and 3rd intervals lasted for 60 and 145 s, respectively. The 2nd interval was a rotation at a shear of 100 1/s for 5 s to deform the sample. The recovery (%) was calculated at a different time within the 3rd interval (2.9, 60, and 145 s) using the following correlation [43].$$\mathrm{Recovery}\;(\%)=\frac{\left(Complex\;viscosity\;of\;the\;ink\;at\;recovery\;(3rd)\;interval\right)}{\left(Complex\;viscosity\;of\;the\;ink\;at\;rest\;\left(1st\right)\;interval\right)}\;\times \;100$$

For all of the inks, an average of three runs of each of the three rheology tests was completed, except for the thixotropy of the HPMC K4 ink due to its liquidity, which caused significant fluctuations in its viscosity.

### 2.5. 3D Printing of the Pills and Films

The pills and films were 3D-printed in the Cellink BioX (San Diego, CA, USA) 3D printer using the design and the developed ink. The printing procedure and printer settings for all of the polymer-based inks were as follows: the prepared ink was loaded into a 3 mL printing cartridge using a spatula. Then, a 410 μm nozzle was attached to the cartridge, and the cartridge was positioned into the printer’s print head. The pill or film was printed at an infill density of 50% with a print speed of 10 mm/s and a printing pressure of 60 kPa. These printing conditions were chosen based on a previous study in our lab [44]. The printing conditions were maintained as constant throughout the entire process during printing. Successfully printed pills and films were dried in an oven (Hermatherm OMS60, Thermo Fischer Scientific, Waltham, MA, US) at 40 °C for 24 h to evaporate any excess solvent. The drying conditions were chosen based on the reported studies, where it was discussed that it did not affect the stability of FNB, as confirmed using DSC and XRD [36,45]. A schematic of the 3D printing process is shown in Figure 4. It is noted that after the ink preparation, in general, 3D printing was carried out. However, making the 3D printing decision after checking the rheology of the ink was found to be the most effective. Figure 5 shows pills and films 3D-printed using all three different polymer (SA, HPC H, and HPMC)-based inks captured using a digital camera.

## 3. Results and Discussion

### 3.1. Viscosity

The impact of the different polymers and their concentrations on the rheological properties (viscosity to the shear rate) of the inks was determined and is presented in Figure 6. For all of the inks, the viscosity decreases as the shear rate increases, known as shear thinning behavior, which is considered suitable for extrusion-based 3D printing techniques [29,46,47,48]. For the SA ink, increasing the content of SA from 0.8% to 1.6% led to a higher viscosity at a low shear rate (Figure 6a). This was due to the rise in the SA content and the decrease in the water content of the ink (Table 1). The findings show that at a low shear rate, inks with a higher water content are less viscous than inks with a lower water content [49]. Both HPC H inks show shear thinning behavior with some variations, unlike the SA ink (Figure 6b). Although the initial viscosity of the 3.2% HPC ink was higher than that of the 4.8% HPC ink at very low shear rates, as the shear increased, the viscosity of the higher-concentration polymeric ink (4.8%) surpassed that of the lower-concentration ink (3.2%), as anticipated. Hence, the effect of the decrease in the water content causes the 4.8% HPC H ink to show a higher viscosity [49,50]. A closer look at the curves in Figure 6b showed that the 4.8% HPC sample caused this initial variation in the trend. This is due to the inconsistent dispersion of the particles within the 4.8% HPC ink, leading to unsteady flow and fluctuations in its rheological properties [51,52]. It is noted that due to clogging, HPC failed to print the correct structure that the SA ink achieved (Figure 5). The polymer HPMC K100 ink showed a higher viscosity than that of the HPMC K4 ink (Figure 6c,d), even though both had a polymer concentration of 1.05%. This is due to the higher chain entanglement of the higher molecular weight of HPMC K100 than the lower molecular weight of HPMC K4 [53,54]. The HPMC-based ink also failed to produce a complete print due to having an insufficient viscosity at low shear rates (Figure 5) [55]. Overall, among the six inks based on three polymers studied, the SA-based ink shows a higher viscosity value with linear shear thinning properties, which is suitable for 3D extrusion-based printing.

### 3.2. Amplitude Sweep

An amplitude sweep test was performed to analyze the viscoelastic behavior of the inks. This test describes the deformation behavior of the inks in the non-destructive deformation range and determines the upper limit of this range [39]. This test’s outcome is represented as amplitude sweep curves consisting of the storage modulus (G′) and loss modulus (G″) vs. the shear strain (%) or shear stress. Figure 7 represents the modulus vs. shear strain curves. The storage modulus represents the elastic (solid-state) portion whereas the loss modulus characterizes the viscous (liquid-state) portion of the viscoelastic behavior of the ink [56]. Each polymeric ink (SA, HPC, and HPMC K100) shows solid-like behavior (Figure 7a–c) except for the HPMC K4 ink (Figure 7d) because for every ink, G′ dominates over G″ up to a certain level of shear strain until it reaches the intersection or crossover point. However, for the HPMC K4 ink, this is flipped, i.e., G″ dominates over G′, and no crossover point is observed. Hence, the HPMC K4 ink shows liquid-like behavior [57,58]. The limit of the LVE region, typically G′, shows a constant plateau value, indicating the range in which the test can be carried out without destroying the internal structure of the samples. At the LVE region, the ink shows solid-like behavior [59,60]. After the crossover point, both the storage and loss moduli are deflected with G″ dominancy, which is the liquid-like structure causing the material to flow [24,61]. This information is useful to know, as it provides an understanding of the printing pressure that is required to be applied to suppress the shear strain in the LVE region (Figure 7a–c) to extrude the ink [46].

A close evaluation of these inks’ amplitude sweep tests was carried out considering the phase shift angle (δ), the yield point (τ_y_), the flow point (τ_f_), and the flow transition index. The ink’s phase shift angle (δ) was calculated considering the following correlation [39].$$\mathrm{Phase}\;\mathrm{shift}\;\mathrm{angle},\;\delta =\tan^{-1}\frac{lossmodulus\;(G\prime \prime )}{storagemodulus\left(G^{\prime }\right)}$$

The modulus values (G′ and G″) were obtained from the LVE region. The yield point is the shear stress value at the limit of the LVE region, whereas the flow point is the shear stress value at the crossover point (G′ = G″). The flow transition index was calculated using the correlation (τ_f_/τ_y_) to evaluate the transition of the ink from the LVE region to show the flow behavior. The ink’s initial structural strength decreases already with predominantly solid-like properties between the yield and flow points. These data are presented in Table 2.

The phase shift angle (δ) is essential for understanding the behavior of the ink at rest. It ranges from 0° to 90°: an ideally elastic behavior δ = 0°, whereas an ideally viscous behavior δ = 90°. If the angle is between 0° and 45°, i.e., 0° ≤ δ < 45°, the samples show a solid or gel-like state, and for a fluid state, the angle is between 45° and 90°, thus 45° < δ ≤ 90° [39,62]. The phase shift angle data in Table 2 confirm that the inks are viscoelastic. The SA, HPC H, and HPMC K100 inks had a δ value less than 45°, indicating that the inks showed solid, gel-like behavior at rest. Only the HPMC K4 ink had a δ value greater than 45°, confirming the fluid-like behavior, which also corroborates Figure 7d in that G″ dominates over G′.

Now, after evaluating the yield and flow points in Table 2, it is shown that all of the inks except the HPMC K4 ink have flow points several times larger than their yield points. The flow transition index is as low as 4.22 to as large as 55.95. No flow point was observed for the HPMC K4 ink, indicating that the ink was liquid across the entire measuring range and confirming the constant physical state of this ink. In an individual comparison of the inks by polymer type, both the 0.8% and 1.6% SA inks, for example, have very close yield point values (5.77 Pa and 2.48 Pa), meaning that there is no significant difference between these two inks. However, when evaluating the flow points, the values of 20.21 Pa and 126.57 Pa differed by a factor of six. This shows that the SA inks needed more pressure to be extruded since these inks had a higher flow point. Due to the higher flow point of the 1.6% SA ink, it had a consistent clogging issue and was riddled with gaps during printing, creating an inconsistent print in both the pill and film dosage forms (Figure 5) [58,63]. These findings indicate that the flow point should be considered when selecting the printing pressure, i.e., the force applied on the ink during printing [64]. A similar observation is found for the HPMC K100 ink. For the HPC H ink, there is no significant difference in the flow points observed. Both formulations (3.2% HPC and 4.8% HPC) show a similar range of values.

### 3.3. The Thixotropy Test

A three-interval thixotropy test was undertaken to determine the post-printing behavior of the inks. In this test, the ink’s time-dependent structural recovery after the application of shear is observed [65]. This also simulates the extrusion of the inks from the cartridge of the printer because when they are subjected to a high shear rate, the inks exhibited a rapid decrease in their viscosity due to their shear thinning behavior. Then, the ink’s ability to restore and return to its original viscosity following the cease of high shear rates indicates the likelihood of the printed filaments spreading post-printing, thereby serving as a measure for the shape fidelity of the printed objects [43]. Figure 8 and Appendix A show the thixotropy profiles (change in viscosity with time) and recovery (%) at different times for all of the polymeric inks studied. During the test, in the first interval, very low shear is applied to simulate the ink’s behavior at rest at low strain within the LVE region. In this rest interval, a breakdown of the initial network structures of the polymer occurs. In the second interval, high shear is applied to simulate the structural breakdown of the sample during extrusion at high strain far beyond the LVE region. Finally, in the third interval, the same low strain value is applied as that in the first interval to simulate the structural recovery of the ink at rest [62]. In 3D printing, when the ink is extruded through the nozzle, the bond between the polymer chains is disrupted due to the shear force [66,67,68]. This stress is released as soon as the ink is extruded, and the internal network bond starts to recover [66]. However, it takes time to recover; in most cases, when the shear is released, a portion of the bonds remain irrecoverable [69]. Hence, a complete recovery may not be observed, i.e., the recovery rate may be less than 100% [46].

An analysis of all of the inks’ thixotropy profiles shows that both the SA and HPC H inks (Figure 8a,b) show a slow recovery with time, indicating that these inks need a long time to recover their initial structures. The HPMC K100 ink shows a complete recovery, whereas the HPMC K4 ink shows fluctuations in its recovery due to its viscosity changing rapidly with time during the measurement (Figure 8c,d). It is noted that the fluctuation observed for HPMC K4 is due to the microstructural changes and competing forces between elastic recovery and viscous flow. The recovery stages of the test involve the reformation of the polymer microstructures, which can lead to fluctuations as the polymer attempts to regain its original properties [70,71,72]. In addition, the recovery process involves both elastic recovery and viscous flow. The interplay between these mechanisms can result in fluctuations as the polymer tries to balance these competing forces [73]. In general, the HPMC K4 ink is considered an ink with low recovery (%), which takes longer to recover, hence being susceptible to spreading during post-printing, resulting in poor shape fidelity [43]. However, an anomaly in our general understanding is observed for the SA-based ink. Despite its poor recovery, the 0.8% SA ink resulted in the best printing (Figure 5). Another observation is that the 1.6% SA and 0.8% SA inks had a similar recovery (%); however, the 1.6% SA ink failed to print well. This anomaly is correlated by closely observing Figure 6a, which shows that at zero shear, the 1.6% ink has a high viscosity and requires a high printing pressure. Hence, using the same printing conditions as those for the 0.8% SA ink cannot provide smooth extrusion of the ink (Figure 5). Overall, all of these findings indicate that not only recovery alone but also comprehensive consideration of all of the rheological characteristics has to be considered to understand the printing outcomes.

### 3.4. Comprehensive Analysis of the SA-, HPC H-, and HPMC K100/K4-Based-Inks’ Rheology

A comprehensive and comparative analysis of all of the polymer-based inks’ rheology and suitability for 3D printing was undertaken. Table 3 and Figure 5 summarize this comparison of the rheological properties of all of the inks investigated and the 3D printing outcomes, respectively. It is shown in Figure 5 that there was no difference in the printing outcomes, even if the pill or the film was printed as the dosage. It was found from the flow tests (Table 3) that all of the inks analyzed showed shear thinning behavior—viscosity decreased as the shear rate increased, which is an essential characteristic needed in the ink for extrusion [41,66,74]. However, viscosity at low shear also needs to be considered because if the inks do not have sufficient viscosity or are too viscous, this causes irregular extrusion or clogging issues (Figure 5) [61]. Hence, eventually, sometimes, the printing process parameters may be required to be adjusted (i.e., for the 0.8% vs. 1.6% SA-based ink), or the polymer compositions need to be adjusted (i.e., for the HPC- and HPMC-based ink).

The amplitude sweep tests showed that all of the inks except the HPMC K4-based ink had higher storage moduli than their loss moduli (Figure 7), a quality that correlates with how well the print can retain its shape [75]. A comparison of the moduli at the flow point (Table 3 and Figure 5) indicates that a higher modulus value is suitable for 3D printing. However, if there is an optimum value or range, it cannot be defined from this limited set of experiments. Another observation from Table 2 was that all of the ink formulations except the HPMC K4-based ink had a sufficient yield and flow stress to avoid unwanted ink dripping from the printing head nozzle before pressure was applied. The shear strain (%) at the flow point (Table 3) supports this. The ink’s behavior, whether it is a viscoelastic solid or liquid, is confirmed by the phase shift angle (δ) (Table 2) and supported by the modulus values (Table 3 and Figure 7). The phase shift angles for every ink except the HPMC K4-based ink tested were below 45°, demonstrating that every ink possessed solid-like qualities [41,62]. The HPMC K4 ink had a higher phase shift angle (68.62°), thus showing liquid-like behavior [62].

We attempted to map the recovery (%) obtained from the thixotropy tests of all of the polymer-based inks, as shown in Table 3. It is found that the SA-based ink has a recovery of <25%, the recovery of the HPC H-based ink is between 25 and 80%, and the HPMC-based ink’s recovery is >80%. Though a higher recovery (%) of the ink is generally expected for good shape fidelity and successful printing outcomes, the opposite was observed [76,77]. The SA-based ink printed successfully, though it had a low recovery (Figure 5). Overall, it is clear that an ink that possesses the rheological characteristics of the 0.8% SA-based ink would be well suited to PAM 3D printing within the printing conditions used for this work. The ink properties favorable for manufacturing 3D-printed structures were shear thinning behavior, high viscosity at low shear, a moderate flow point, a low shift phase angle, slow recovery, and a low recovery percentage.

## 4. Conclusions

Several inks were prepared using the commonly used polymers SA, HPC H, HPMC K100, and HPMC K4 and evaluated for extrusion-based PAM-type 3D printing of pharmaceuticals by investigating their rheology. The inks included polymers, various functional excipients, the model drug FNB, and a solvent (water or a water–ethanol mixture). A pill or film, as an oral drug product dosage form, was printed using the 3D printer, and the printing outcomes were analyzed and provisionally connected with the inks’ rheological properties, which were measured using an Anton Paar MCR 302 rheometer. This study considered several rheological properties of the inks, including the viscosity, storage and loss moduli, yield point, flow point, flow behavior, phase shift angle, and recovery (%) at different times.

This study found that only the 0.8% SA ink was suitable for 3D-printing the pills and films, with necessary properties such as shear thinning behavior, high viscosity at rest, a high modulus at the flow point, a low recovery ratio, and slow recovery. The other inks require improvements in their rheological properties. This study also found that (i) the printing process parameters may need to be adjusted based on the ink’s rheological properties, especially the ink’s flow point and (ii) inks can be used to print various dosage forms as long as they have the proper rheological properties and the printing process has the optimum operating conditions. This research is helpful in that through its rheological evaluation, an ink suitable for extrusion-based 3D printing of pharmaceuticals can now be developed more conveniently and efficiently.

## Figures and Tables

**Figure 1 micromachines-16-00163-f001:**
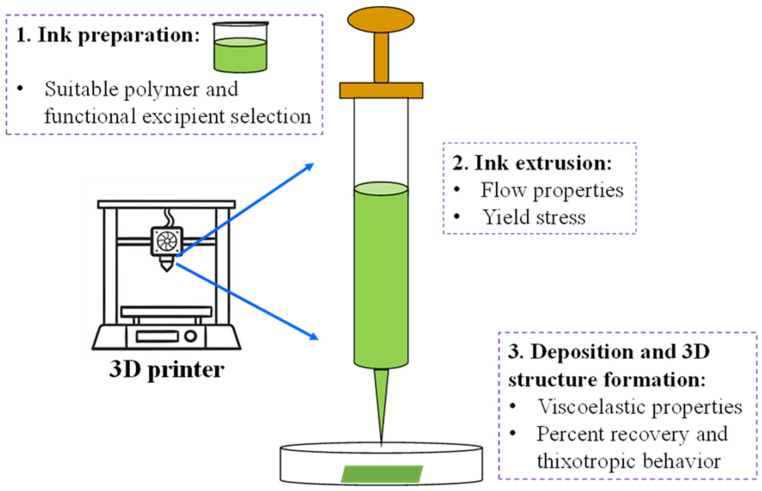
Schematic presentation of the steps involved in the pressure-assisted microsyringe (PAM)-type extrusion-based 3D printing process.

**Figure 2 micromachines-16-00163-f002:**
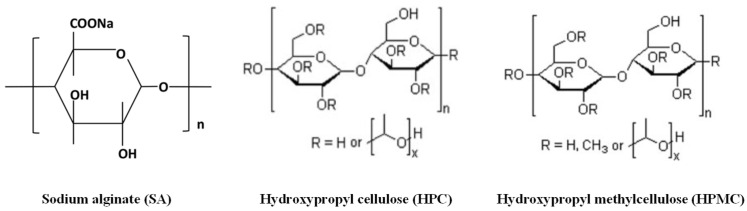
Molecular structure of sodium alginate (SA), hydroxypropyl cellulose (HPC), and hydroxypropyl methylcellulose (HPMC) [34,35].

**Figure 3 micromachines-16-00163-f003:**
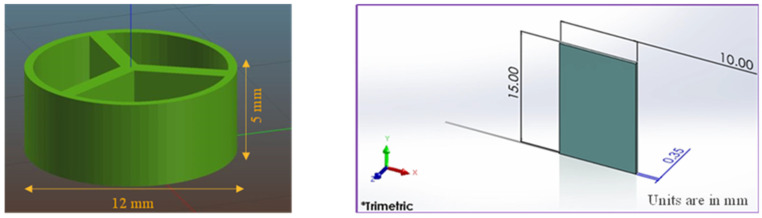
Design of pill and thin film used for 3D printing.

**Figure 4 micromachines-16-00163-f004:**
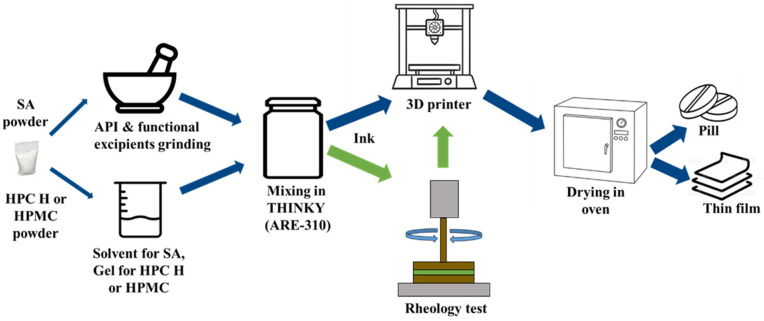
Schematic presentation of the 3D printing of pills and thin films.

**Figure 5 micromachines-16-00163-f005:**
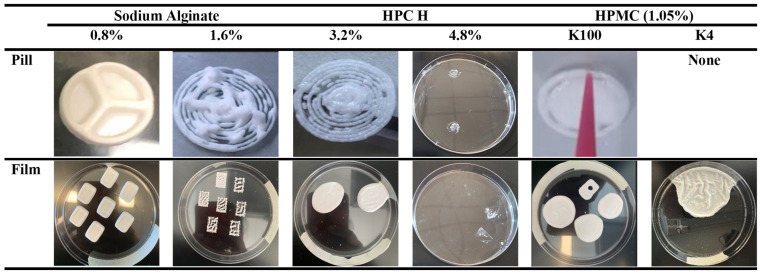
Three-dimensional printing outcomes using various polymer-based inks. The printing conditions were a 50% infill density, a 10 mm/s print speed, and a 60 kPa printing pressure.

**Figure 6 micromachines-16-00163-f006:**
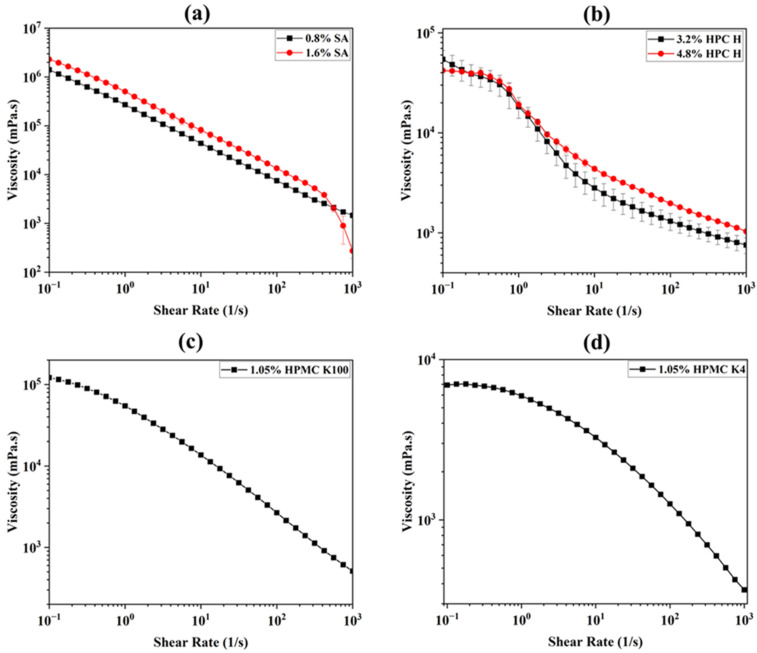
Viscosity curves of (**a**) sodium alginate-, (**b**) HPC H-, and (**c**,**d**) HPMC K100/K4-based ink. The tests were performed at 25 °C, a 1 Hz frequency, and a 0.1 to 1000 1/s shear rate.

**Figure 7 micromachines-16-00163-f007:**
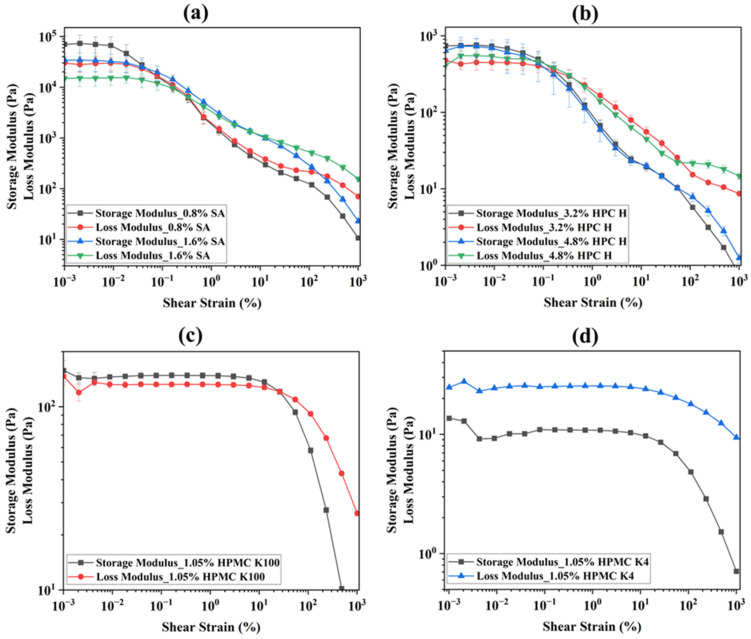
Amplitude sweep curves (modulus vs. shear strain) of (**a**) sodium alginate-, (**b**) HPC H-, and (**c**,**d**) HPMC K100/K4-based ink. The tests were performed at 25 °C, a 1 Hz frequency, and a 0.001 to 1000%. shear strain.

**Figure 8 micromachines-16-00163-f008:**
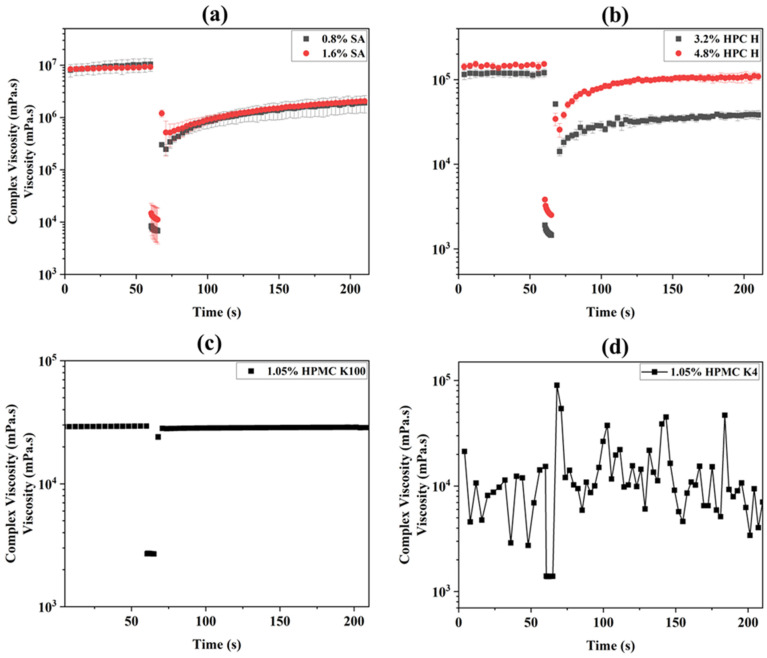
Thixotropy profiles of (**a**) sodium alginate-, (**b**) HPC H-, (**c**,**d**) HPMC K100/K4-based ink.

**Table 1 micromachines-16-00163-t001:** The formulation used to prepare the ink.

Polymer Name	Polymer		Drug	Functional Excipient				Solvent	
			FNB	Mannitol	PEG	PVP	SDS	Ethanol	Water
	(%, *w*/*w*)	(g)	(g)	(g)	(g)	(g)	(g)	(g)	(g)
Sodium Alginate (SA)	0.80	0.24	6.0	7.50	3	1.5	-	-	11.76
	1.60	0.48	6.0	7.50	3	1.5	-	-	11.52
HPC H	3.20	0.96	6.0	7.50	3	1.5	-	-	11.04
	4.80	1.44	6.0	7.50	3	1.5	-	-	10.56
HPMC K100	1.05	0.32	1.5	7.35	-	-	0.15	5.25	15.43
HPMC K4	1.05	0.32	1.5	7.35	-	-	0.15	5.25	15.43

**Table 2 micromachines-16-00163-t002:** Amplitude sweep test data (average ± SD) on various polymer-based inks.

Polymer and Its Concentration (%, *w*/*w*)	Phase Shift Angle, δ	Yield Point or Stress, τ_y_	Flow Point or Stress, τ_f_	Flow Transition Index, τ_f_/τ_y_
	(°)	(Pa)	(Pa)	
0.8% SA	23.61 ± 2.91	5.77 ± 3.27	20.21 ± 5.70	4.22 ± 1.78
1.6% SA	25.41 ± 3.42	2.48 ± 0.65	126.57 ± 19.57	55.95 ± 19.19
3.2% HPC H	30.62 ± 0.80	0.09 ± 0.04	0.83 ± 0.37	9.78 ± 1.33
4.8% HPC H	37.10 ± 3.19	0.10 ± 0.08	0.53 ± 0.58	5.44 ± 3.20
1.05% HPMC K100	41.74 ± 0.18	1.39 ± 0.05	45.30 ± 1.50	32.66 ± 0.11
1.05% HPMC K4	68.62 ± 3.90	-	-	-

**Table 3 micromachines-16-00163-t003:** Summary of rheological property investigation and suitability for 3D printing of various polymer-based inks.

Test Name	Parameter	Polymer					
		Sodium Alginate		HPC H		HPMC (1.05%)	
		0.8%	1.6%	3.2%	4.8%	K100	K4
**Rheological Properties (Average ±SD)**							
**Flow test**	*Viscosity (mPa.s) at a 0.1 shear rate (1/s)*	1.40 × 10^6^ ± 1.14 × 10^5^	2.30 × 10^6^ ± 7.88 × 10^4^	5.44 × 10^4^ ± 1.21 × 10^4^	4.20 × 10^4^ ± 1.99 × 10^3^	1.22 × 10^5^ ± 2.40 × 10^3^	6.92 × 10^3^ ± 1.20 × 10^2^
	*Shear thinning behavior*	Linear shear thinning		Shear thinning			
**Amplitude sweep**	*Modulus at flow point (Pa)*	1.52 × 10^4^ ± 9.52 × 10^3^	1.30 × 10^3^ ± 1.95 × 10^2^	3.43 × 10^2^ ± 8.80 × 10^1^	4.35 × 10^2^ ± 8.42 × 10^1^	1.21 × 10^2^ ± 4.28 × 10^0^	-
	*Shear strain at flow point (%)*	0.27 ± 0.30	6.98 ± 1.11	0.20 ± 0.14	0.09 ± 0.10	26.5 ± 0.00	-
	*Ink behavior*	Viscoelastic					Viscoelastic liquid
**Thixotropy**	*Recovery (%)*	<25		25–80		>80	-
**3D printing outcomes**							
	*Shape fidelity*	Successful	Unsuccessful				

## Data Availability

Additional data will be made available on request.

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
