# Peer review of "Extrusion-Based 3D Printing of Pharmaceuticals—Evaluating Polymer (Sodium Alginate, HPC, HPMC)-Based Ink’s Suitability by Investigating Rheology"

_micromachines, 2025, doi:10.3390/mi16020163_

Round 1

Reviewer 1 Report

Comments and Suggestions for Authors

This research investigated the rheology of polymer-based inks (sodium alginate, HPC, HPMC) for PAM 3D printing in pharmaceuticals. It evaluated printability of pills and films, finding 0.8% SA ink suitable and highlighting key rheological factors for ink selection. This work is interesting, but I have several concerns. Here are my comments:

In the statistical analysis, it should be clearly stated whether the differences in rheological properties between different inks are statistically significant. For example, when comparing the viscosities of different inks at different shear rates, a statistical test (such as ANOVA) need be used to determine if the differences are significant.

Why were 0.8% and 1.6% SA selected?

The paper should be more practical by providing more guidance on how to adjust the ink formulations or printing parameters in a real manufacturing setting. For example, suggesting possible ranges of polymer concentrations or printing pressures that could be used to optimize the printing process based on the observed rheological properties.

This paper lacks a recent related advancements, like Multimodal Strain Sensing System for Shape Recognition of Tensegrity Structures by Combining Traditional Regression and Deep Learning Approaches; and Predicting flow status of a flexible rectifier using cognitive computing.

You mentioned that some inks showed fluctuations in certain tests (e.g., HPC H inks in the shear thinning behavior and HPMC K4 in the thixotropy test). The possible causes of these fluctuations should be discussed and how they might have affected the overall reliability of your results?

Comments on the Quality of English Language

In the results section, it might be better to group the discussion of each polymer's properties more tightly. For example, when discussing the viscosity of all inks, instead of jumping back and forth between different polymers, present all the viscosity data and analysis for one polymer before moving on to the next. This would make it easier for the reader to follow the comparison.

Author Response

Reviewer #1

Comments:

Comments and Suggestions for Authors

This research investigated the rheology of polymer-based inks (sodium alginate, HPC, HPMC) for PAM 3D printing in pharmaceuticals. It evaluated printability of pills and films, finding 0.8% SA ink suitable and highlighting key rheological factors for ink selection. This work is interesting, but I have several concerns. Here are my comments:

  1. In the statistical analysis, it should be clearly stated whether the differences in rheological properties between different inks are statistically significant. For example, when comparing the viscosities of different inks at different shear rates, a statistical test (such as ANOVA) need be used to determine if the differences are significant.

Authors: Thanks for your excellent comment. In this research, we showed that to achieve a successful 3D printed dosage, it is essential to consider all rheological properties (i.e., viscosity, modulus, or thixotropy). We find that printability is independent of the dosage form. An optimal ink is viscoelastic solids with specific rheological characteristics. Hence, we connected the ink suitability for extrusion-based 3D printing by investigating the rheology. In this study, we did not focus on using statistical tools (Analysis of Variance) to analyze the significance of the different ink’s rheological characteristics.

  1. Why were 0.8% and 1.6% SA selected?

Authors: Thanks for another excellent comment. In our lab, a brief preliminary study [1] was done to make inks using different percentages of polymers (e.g., sodium alginate) and evaluated their printability. From that brief study, it was found that 0.8% sodium alginate gives good printability. Subsequently, it was chosen for this study. Then, the concentration was increased to double to see the effect on printability and rheology. A sentence with the reference has been added in section 2.2.

Reference:

  1. Kimbell, G. Rheology Impacts on Extrusion-Based 3D Printing of Polymeric Structures Incorporating Pharmaceuticals. M.S., North Carolina Agricultural and Technical State University: United States -- North Carolina, 2020.

  1. The paper should be more practical by providing more guidance on how to adjust the ink formulations or printing parameters in a real manufacturing setting. For example, suggesting possible ranges of polymer concentrations or printing pressures that could be used to optimize the printing process based on the observed rheological properties.

Authors: Thank you for your excellent comment. This study explains how rheology can guide us to the proper polymer-based ink selection suitable for extrusion-based 3D printing. In this work, we selected the polymer concentration and printing conditions based on our previous brief preliminary studies, where we identified the optimal printing conditions [1] and the polymer concentration [2] that work best for 3D printing. We have updated this information in Section 2.5 and Section 2.2 with the references. We view the reviewer's comments as a direction for future research to further expand on the findings of our current study.

References:

  1. Olawuni, D. Optimization of Process Parameters for Extrusion-Based 3D Printing of Polymeric Structures Incorporating Pharmaceuticals. M.S., North Carolina Agricultural and Technical State University: United States -- North Carolina.
  2. Kimbell, G. Rheology Impacts on Extrusion-Based 3D Printing of Polymeric Structures Incorporating Pharmaceuticals. M.S., North Carolina Agricultural and Technical State University: United States -- North Carolina, 2020.

  1. This paper lacks a recent related advancements, like Multimodal Strain Sensing System for Shape Recognition of Tensegrity Structures by Combining Traditional Regression and Deep Learning Approaches; and Predicting flow status of a flexible rectifier using cognitive computing.

Authors: Thank you for another excellent comment. This study explains how rheology can guide us to the proper polymer-based ink selection suitable for extrusion-based 3D printing. The printing outcome was captured using a digital camera. The study does not consider predictability or shape recognition, so artificial intelligence tools were not used in this study. Hence, the information related to those areas is not considered.

  1. You mentioned that some inks showed fluctuations in certain tests (e.g., HPC H inks in the shear thinning behavior and HPMC K4 in the thixotropy test). The possible causes of these fluctuations should be discussed and how they might have affected the overall reliability of your results?

Authors: Thanks for this excellent comment. Comparing both 3.2 % and 4.8% HPC H inks showed that at a very low shear rate, the initial viscosity of the 3.2% HPC ink was higher than that of the 4.8% HPC ink. As shear increased, the viscosity of the higher-concentration polymeric ink (4.8%) surpassed that of the lower-concentration ink (3.2%), as anticipated. A close look at the curves in Figure 6b showed that the 4.8% HPC sample caused this initial variation in the trend. This is due to the inconsistent dispersion of particles within the 4.8% HPC ink, leading to unsteady flow and fluctuations in rheological properties [1, 2]. It is found that 4.8% HPC ink exhibited particle agglomeration, causing the nozzle to clog frequently and preventing smooth ink flow. On the other hand, the fluctuation observed for HPMC K4 in the thixotropy test is due to the microstructural changes and competing forces between elastic recovery and viscous flow. The recovery stages of the test involve the reformation of polymer microstructures, which can lead to fluctuations as the polymer attempts to regain its original properties [3-5]. In addition, the recovery process involves both elastic recovery and viscous flow. The interplay between these mechanisms can result in fluctuations as the polymer tries to balance these competing forces [6]. A brief explanation of the causes of the fluctuations in both scenarios has been added to the manuscript. The fluctuations observed for these two inks do not impact the overall findings of this research study.

References:

  1. Guo, Z., Fei, F., Song, X., & Zhou, C. (2022, June). Analytical study of shear-thinning fluid flow in direct ink writing process. In International Manufacturing Science and Engineering Conference (Vol. 85802, p. V001T01A034). American Society of Mechanical Engineers.
  2. Guo, Z., Fei, F., Song, X., and Zhou, C. (March 15, 2023). "Analytical Study and Experimental Verification of Shear-Thinning Ink Flow in Direct Ink Writing Process." ASME. J. Manuf. Sci. Eng. July 2023; 145(7): 071001. https://doi.org/10.1115/1.4056926
  3. Bercea, M. (2023). Rheology as a tool for fine-tuning the properties of printable bioinspired gels. Molecules, 28(6), 2766.
  4. Barrulas, R. V., & Corvo, M. C. (2023). Rheology in product development: an insight into 3D printing of hydrogels and aerogels. Gels, 9(12), 986.
  5. Agrawal, R., & García-Tuñón, E. (2024). Interplay between yielding,‘recovery’, and strength of yield stress fluids for direct ink writing: new insights from oscillatory rheology. Soft Matter, 20(37), 7429-7447.
  6. Menard, K. P., & Menard, N. R. (2020). Rheology Basic: Creep-Recovery and Stress Relaxation. Dynamic Mechanical Analysis; CRC Press: Boca Raton, FL, USA, 45-68.

  1. Comments on the Quality of English Language

In the results section, it might be better to group the discussion of each polymer's properties more tightly. For example, when discussing the viscosity of all inks, instead of jumping back and forth between different polymers, present all the viscosity data and analysis for one polymer before moving on to the next. This would make it easier for the reader to follow the comparison.

Authors: Thank you very much for your valuable comment. In the results and discussion section, each ink rheology was discussed, compared with other inks, and eventually connected with printing outcomes. We have revised the results and discussion section and made the necessary changes. We also revised the manuscript to improve the quality of the English.

Reviewer 2 Report

Comments and Suggestions for Authors

The manuscript investigates the effect of the ink's rheological properties on PAM printing by varying the polymers used in the inks. This approach could be valuable for optimizing inks for PAM 3D printing in pharmaceutical applications. I recommend the publication of this manuscript in Micromachines following minor revisions. Below are some suggestions:

1.      It would be helpful to include a scheme with the structures of SA, HPC, and HPMC in the main text for better clarity.

2.      The polymer concentrations in the different inks vary, e.g., 0.8%-1.6% for SA, 3.2%-4.8% for HPC H, and 1.05% for HPMC. Some rationale should be provided to explain the choice of specific polymer concentrations.

3.      The resolution of the images in Figure 4 could be improved for better quality.

Author Response

Reviewer #2

Comments:

Comments and Suggestions for Authors

The manuscript investigates the effect of the ink's rheological properties on PAM printing by varying the polymers used in the inks. This approach could be valuable for optimizing inks for PAM 3D printing in pharmaceutical applications. I recommend the publication of this manuscript in Micromachines following minor revisions. Below are some suggestions:

Authors: Thanks for your positive feedback. We have revised the manuscript.

  1. It would be helpful to include a scheme with the structures of SA, HPC, and HPMC in the main text for better clarity.

Authors: Thanks for your comment. We added the molecular structure of SA, HPC, and HPMC polymers in Figure 2.

  1. The polymer concentrations in the different inks vary, e.g., 0.8%-1.6% for SA, 3.2%-4.8% for HPC H, and 1.05% for HPMC. Some rationale should be provided to explain the choice of specific polymer concentrations.

Authors: Thanks for this constructive comment. In our lab, a brief preliminary study [1] was done to make inks using different percentages of polymers (e.g., sodium alginate) and evaluated their printability. That brief study guides us to select the polymer concentration for this study. A sentence with the reference has been added in section 2.2.

Reference:

  1. Kimbell, G. Rheology Impacts on Extrusion-Based 3D Printing of Polymeric Structures Incorporating Pharmaceuticals. M.S., North Carolina Agricultural and Technical State University: United States -- North Carolina, 2020.
  2. The resolution of the images in Figure 4 could be improved for better quality.

Authors: Thanks for another excellent comment. The images were captured using a digital camera. Due to background reflection, printing outcomes pictures were not clarified for only 4.8% of HPC. Hence, we replaced the images with new images in the figure.

Reviewer 3 Report

Comments and Suggestions for Authors

In this manuscript, the study investigated the polymer-based composite inks suitable for drug printing in Pressure-Assisted Microsyringe (PAM) 3D printing, focusing on the performance of inks based on different polymers (sodium alginate, HPC H, HPMC K100, and K4) during the printing process. Overall, this is a meaningful work with reasonable theoretical analysis and experimental verifications. However, several points must be carefully addressed before the paper can be accepted for publication.

1.The paper explores the development of polymer-based inks for Pressure-Assisted Microsyringe (PAM) 3D printing, focusing on their rheological properties and printing performance. Could the authors elaborate on the novelty of their approach compared to existing research on 3D printing inks?

2.The manuscript investigates the rheological properties of polymer-based inks for Pressure-Assisted Microsyringe (PAM) 3D printing. It would be valuable if the authors could provide a detailed comparison of ink performance under varying conditions of pressure, speed, and temperature. How do these factors influence the stability and applicability of the inks across different printing scenarios?

3.Could the authors provide more detailed annotations in Figures 5 and 6, such as specifying the printing conditions and concentrations corresponding to each curve? Additionally, could the authors include brief textual explanations of key trends alongside the figures to make them easier for readers to understand?

Author Response

Reviewer #3

Comments:

Comments and Suggestions for Authors

In this manuscript, the study investigated the polymer-based composite inks suitable for drug printing in Pressure-Assisted Microsyringe (PAM) 3D printing, focusing on the performance of inks based on different polymers (sodium alginate, HPC H, HPMC K100, and K4) during the printing process. Overall, this is a meaningful work with reasonable theoretical analysis and experimental verifications. However, several points must be carefully addressed before the paper can be accepted for publication.

Authors: Thanks for your positive feedback. We have revised the manuscript.

  1. The paper explores the development of polymer-based inks for Pressure-Assisted Microsyringe (PAM) 3D printing, focusing on their rheological properties and printing performance. Could the authors elaborate on the novelty of their approach compared to existing research on 3D printing inks?

Authors: Thanks for this excellent comment. Developing a polymer-based ink suitable for PAM-type 3D printing requires trial and error, which is time-consuming and resource-intensive, especially for high-value products such as pharmaceuticals. If the polymer-based ink does not have the proper rheological characteristics, it may create problems such as clogging in the print head nozzle or defective prints. Polymers Sodium alginate (SA), Hydroxypropyl cellulose (HPC), and Hydroxy-propyl methylcellulose (HPMC) are commonly used in the pharmaceutical industry. Literature was found that reported ink preparation using these polymers. However, no study has evaluated all three of these polymers, compared their different polymeric ink’s rheological properties, and eventually connected them with the 3D printing outcomes for various oral dosages.  

This research study aims (i) to assess the inks prepared using polymer SA, HPC H, or HPMC K100 or K4 by investigating and comparing their rheology and to connect the rheological data with the ink’s applicability for the PAM type 3D printing; (ii) to observe if there are any variations in printing outcomes due to the dosage variations such as pills and films, most commonly used in an oral dosage form. The novelty of this research work is exploring these aims. A sentence has been added in the Introduction to mention the novelty of the work.

  1. The manuscript investigates the rheological properties of polymer-based inks for Pressure-Assisted Microsyringe (PAM) 3D printing. It would be valuable if the authors could provide a detailed comparison of ink performance under varying conditions of pressure, speed, and temperature. How do these factors influence the stability and applicability of the inks across different printing scenarios?

Authors: Thanks for this constructive comment. In our study, we did not vary processing parameters, though this does impact printability. We chose the printing parameters based on the previous research from our lab and kept the condition constant [1]. We have updated this information in Section 2.5. It is also noted that no stability of the inks was studied in this study. Hence, we view the reviewer's comments as a direction for future research.

Reference:

  1. Olawuni, D. Optimization of Process Parameters for Extrusion-Based 3D Printing of Polymeric Structures Incorporating Pharmaceuticals. M.S., North Carolina Agricultural and Technical State University: United States -- North Carolina.

  1. Could the authors provide more detailed annotations in Figures 5 and 6, such as specifying the printing conditions and concentrations corresponding to each curve? Additionally, could the authors include brief textual explanations of key trends alongside the figures to make them easier for readers to understand?

Authors: Thank you for another excellent comment. Figures 5 and 6 have now been updated to Figures 6 and 7. We have provided printing and testing conditions in Figures 5, 6, and 7. To keep the figures visually clear, we did not add additional text to explain the key trends. A detailed explanation of each of these figures is given in sections 3.1-3.4.

Round 2

Reviewer 1 Report

Comments and Suggestions for Authors

adequate